# A Cone Beam CT Study on the Correlation between Crestal Bone Loss and Periapical Disease

**DOI:** 10.3390/jcm12062423

**Published:** 2023-03-21

**Authors:** Sari A. Mahasneh, Abeer Al-Hadidi, Fouad Kadim Wahab, Faleh A. Sawair, Mohammad Abdalla AL-Rabab’ah, Sarah Al-Nazer, Yara Bakain, Cosimo Nardi, Joanne Cunliffe

**Affiliations:** 1School of Dentistry, University of Jordan, Amman 11942, Jordan; 2Division of Dentistry, School of Medical Sciences, University of Manchester, Manchester M13 9PL, UK; 3Independent Researcher, Amman 11931, Jordan; 4Department of Experimental and Clinical Biomedical Sciences, Radiodiagnostic Unit n. 2, University of Florence—Azienda Ospedaliero-Universitaria Careggi, 50134 Florence, Italy

**Keywords:** bone loss, cone-beam CT, apical periodontitis, endodontics

## Abstract

The aim of this study was to determine whether the degree of bone loss around teeth can be linked to the loss of vitality of adjacent teeth and periapical disease, which necessitates root canal treatments. Three hundred and twenty-one full maxilla cone-beam computed tomography scans were examined. The parameters investigated included the degree of crestal bone loss in relation to the cementoenamel junction, the presence/absence of apical periodontitis, and the presence/absence of root canal treatments. Out of the 2001 teeth examined, 696 (34.8%) showed evidence of crestal bone loss. The degree of crestal bone loss was classified as mild, moderate, or severe. A significant association (*p* < 0.001) was found between the presence of crestal bone loss around a tooth and root canal treatment of that tooth. It was found that it is more likely for teeth with crestal bone loss to be root canal treated compared to teeth with existing root canal treatment and healthy crestal bone levels. Furthermore, teeth with buccal or lingual crestal bone loss were significantly associated with a higher rate of periapical disease than teeth without crestal bone loss (*p* < 0.001). CBCT identified the severity of bone loss on all surfaces of the teeth, and the most common presentation was bone loss to the mid-root level. Teeth with crestal bone loss were significantly more likely to be associated with a higher rate of periapical disease. Teeth with crestal bone loss were more likely to be root treated than teeth with healthy crestal bone levels.

## 1. Introduction

The relationship between endodontic and periodontal diseases has always been a topic of interest since it was first discussed by Simring and Goldberg [1]. The relationship between pulpal tissue and the tooth-supporting tissues (i.e., periodontal tissues) stems from the fact that they share the same embryologic origin, which is the ectomesenchyme. The dental papilla results in the development of the pulp, while the dental follicle forms the periodontal supporting tissues. The spread of pathology across tissues (i.e., the periodontium and the pulp) was first described as “retrograde periodontitis” by Simring and Goldberg [1], where pulpal disease spreads from the root canal systems to the surrounding periodontal tissues through the apical foramen or lateral canals, causing periodontal disease.

Rotstein and Simon [2] proposed categorizing the aetiology of the disease into two main categories: microorganisms and non-infectious factors. The microbial factors include the commonly well-known periodontal diseases (i.e., gingivitis and periodontitis) and endodontic diseases (i.e., irreversible pulpitis and apical periodontitis). These diseases are caused by microorganisms, mainly bacteria, that induce damage to the infected tissue (e.g., pulpal tissue) and might spread to the adjacent tissues, especially to the periodontal attachment through lateral or accessory canals, and vice versa. The other aetiology subcategory is caused by non-infective factors such as trauma, perforations, or resorptions. These two factors share the same pathway for the spread of the disease by creating a path of communication between periodontal and pulpal tissues, thus facilitating the spread of the disease (3).

The main aim of the root canal treatment is to clean the root canal system and lower the microbial load, followed by obturating the root canals, which should provide a hermetic seal. The concept behind providing this seal is to entomb the remaining microbes within the root canal system and prevent the reinfection of this very complex anatomical structure. The seal can be divided into two main parts, the coronal seal and the apical seal. The coronal seal relies on a sound definitive restoration with intact margins that would prevent microbes from accessing the root canal system coming from the oral cavity. While for the apical seal, it is mainly provided by filling the root canal system with a suitable material, usually gutta percha with sealer. This hermetic apical seal will be responsible for preventing the microbes and their endotoxins from accessing the periodontal tissues surrounding them.

The controversial question has always been whether the coronal seal is of more importance than the apical seal or the opposite. In 1995, Ray and Trope [3] looked at 1010 endodontically treated teeth restored with a definitive restoration, grading the technical quality of the coronal restoration and the root canal filling using the absence of apical inflammation as an indication of disease presence. It was concluded that the presence of a good restoration (80%) significantly increases the percentage of cases with an absence of apical inflammation when compared to good endodontic treatment (75.7%).

The controversy was noted when Tronstad [4] repeated the study in 2000, using the same methodology as Ray and Trope [3]. Tronstad [4] found a different result, noting that the groups with good endodontic treatment had the highest success rate. Being 81% when combined with a good restoration and 71% when combined with a poor restoration. Combining the results of both studies [3,4], it is evident that a combination of a good definitive coronal restoration with a good root canal treatment results in the highest success rate.

The studies [3,4] discuss the importance of having a good seal on the root canal system to prevent the access of the microbes and their endotoxins to the surrounding tissues. However, these studies [3,4] talked about the major entrances and exits of the root canal system and did not discuss sealing the lateral or accessory canals of the root canal system, nor did they discuss the potential pathway that dentinal tubules can provide into or out of the root canal system.

In 1975, De Deus [5] examined the distribution of the accessory canals and found that 17% of them are located apically, 9% in the middle third, and 2% in the coronal third of the main canal. Kim [6] later confirmed this finding in a scanning electron microscope study, which showed that 98% of the ramifications and canal complexities exist in the apical 3 mm of the root. The treatment of endo-periodontal diseases can cause a dilemma for clinicians on where to start and the sequence of treatment. The best way to treat a disease is to correctly identify its etiology. Various classifications have been proposed, with the very first one being introduced by Simon et al. [7]. In this classification, periodontal disease was classified into five main subcategories, with each disease being considered primary or secondary. Simon et al.’s [7] classification categories were “periodontal lesions”, “endodontic lesions”, “periodontal lesions with secondary endodontic involvement”, “endodontic lesions with secondary periodontal involvement”, and “true combined lesions”.

The consensus report of the 2017 World Workshop on the Classification of Periodontal and Peri-Implant Diseases and Conditions [8] classified endo-periodontal disease into two categories depending on whether there was root damage or not. The two categories were subdivided into three grades each. The classification is listed in Table 1.

Radiographic assessment can be helpful to have a general insight into the bone structures and apical status, but only mesial and distal levels can be assessed due to the 2-dimensional imaging of 3-dimensional structures, anatomic noise, superimposition, and geometric distortion [9,10,11]. With the introduction of 3-dimensional (3D) imaging techniques, especially cone beam computed tomography (CBCT), the relationship between bone levels and roots can be clearly assessed with no superimposition, in addition to its ability to assess buccal and palatal/lingual bone levels [12]. The CBCT imaging technique showed excellent accuracy when diagnosing apical periodontitis [12]. The accuracy of CBCT was superior to that of intraoral digital radiography for measuring periodontal bone loss, including horizontal, one-, two-, and three-wall defects, craters, dehiscence, and fenestrations [13]. Another systematic review showed that CBCT imaging had high accuracy when visualizing periodontal structure, in particular around the maxillary molar, where there tends to be complex geometry [14].

To our knowledge, no previous paper has correlated the degree of bone loss with the prevalence of apical periodontitis using CBCT. The aim of this retrospective study was to determine whether the degree of bone loss can be linked to the loss of vitality of the tooth and its effect on the development of periapical disease, which necessitates root canal treatment. CBCT imaging was used as the reference standard to assess the bone loss.

## 2. Materials and Methods

Ethical approval for this study was granted by the ethical committee at the school of dentistry at the Jordan University (reference IRB 234/2020). Four hundred twenty-one CBCT examinations were retrieved from October 7, 2019, to February 22, 2020. The scans were identified from the radiology department logbook for patients analyzed by different specialty clinics at the Jordan University Hospital (Amman, Jordan). Scans were justified as most of the included patients had them for implant planning reasons, impacted teeth, or endodontic reasons. Scans included are needed to have a full view of the maxilla with at least one permanent maxillary first molar. Scans with poor imaging technique, metal artifacts, or any abnormalities that limited the visualization of teeth and bone were excluded. Scans suffering from motion artifacts and patients with focal lesions or metabolic processes in the maxilla were also excluded [15].

The methodology of this study was planned to be in accordance with Mol and Balasundaram [16], where applicable. Two qualified general dentists were trained by an oral and maxillofacial radiologist to read CBCT images and identify the parameters investigated, including the degree of bone loss in relation to the cementoenamel junction (CEJ) and the presence/absence of apical periodontitis. A sample of twenty scans that were not included in the study were used for training the readers. The results were then discussed with the radiologist, and further training was undertaken 2 weeks later to ensure consistency and give intra-examiner feedback. There was high agreement when Kappa statistical analysis was undertaken, with an agreement value of 80% (*p* < 0.001). The data extraction was performed under standardized viewing conditions in a low-lit room using Carestream^®^ 3D imaging software that allows brightness and contrast adjustments. All CBCT scans were analyzed on the same radiology department workstation to exclude any screen bias. The extracted data consisted of the patient’s gender, age, type, and location of teeth. In addition, the degree of bone loss and the presence/absence of apical periodontitis at any of the roots were also analyzed. An axial CBCT cut was used for the detection of apical disease, while active scrolling through labio-palatal sections was used to detect any loss in the alveolar bone levels surrounding the tooth. The presence of root canal treatment was also noted. The data collected was classified as nominal in terms of whether there was apical pathosis or not and categorical regarding the amount of bone loss, whether it involved the coronal, middle, or apical third of the tooth. A periapical lesion was defined as a periapical radiolucent area that was in contact with the radiographic apex of the root and measured at least twice the width of the periodontal ligament space [17]. The distance between the cementoenamel junction and the alveolar crest was measured for each site. The amount of bone loss was used to assess the severity of periodontitis. For simplification, the root involved was divided into three equal thirds from the apex to the CEJ, and then bone loss was assessed by relating the level of the crestal bone to the root level. Therefore, bone loss limited to the cervical part of the root was considered mild, while bone loss extending to the middle third of the root was considered moderate and severe if bone loss reached the apical third of the root. Data was entered into statistical data analysis software (IBM SPSS Data Editor).

Statistical analysis was performed using SPSS for Windows release 16.0 (SPSS Inc., Chicago, IL, USA). Descriptive statistics were generated, and the Chi-square test was used to examine differences between groups. The significance level was stated as a *p*-value < 0.05.

## 3. Results

A total of 321 patients (176 females and 145 males) met our inclusion criteria. There were 2001 teeth examined: 468 first premolars, 455 second premolars, 488 first molars, 467 second molars, and 123 third molars. Of the 2001 teeth examined, 696 (34.8%) showed evidence of buccal or lingual bone loss. The degree of periodontitis was mild in 272 teeth (39.1%), moderate in 296 (42.5%), and severe in 128 teeth (18.4%) (Figure 1).

The periapical regions of teeth were examined, and 688 (34.4%) had apical periodontitis. Among the teeth with periapical lesions, 38 were third molars; 147 were second molars; 185 were first molars; 144 were second premolars; and 174 were first premolars. There were 302 teeth (15.1%) that had been root canal treated.

A significant association (*p* < 0.001) was found between the presence of periodontal bone loss around a tooth and root canal treatment of the same tooth. Root canal treatment (RCT) was observed in 20.1% of teeth with signs of crestal bone loss and 12.4% of teeth with no signs of crestal bone loss. There was no significant association between the degree of periodontal bone loss—mild, moderate, or severe—and the prevalence of RCT (*p* = 0.166). Significant differences (*p* = 0.007) in the prevalence of RCT were found when the different types of teeth were considered, as shown in Figure 2.

The presence of RCT was also affected by whether the tooth with crestal bone loss was single- or multi-rooted (*p* < 0.001). Of the single-rooted teeth with periodontitis (87 teeth), 36.8% had RCT, compared to 17.7% of the multi-rooted teeth (609 teeth).

Teeth with crestal bone loss were significantly associated with a higher rate of periapical pathology (51.0%) compared to a rate of (25.5%) when no crestal bone loss was present (*p* < 0.001). There was no significant association between the degree of crestal bone loss and the prevalence of periapical periodontitis (*p* = 0.865). In addition, no significant difference in the prevalence of periapical lesions was found when different types of teeth—premolars and molars—with crestal bone loss were considered (*p* = 0.480). The presence of periapical periodontitis was affected by whether the tooth with crestal bone loss was single (8.9%) or multi-rooted (13.1%) (*p* < 0.001). It was also noted that teeth with moderate or severe bone loss tended to be associated with a higher prevalence of a periapical lesion (*p* < 0.001), irrespective of whether the tooth underwent root canal treatment or not.

## 4. Discussion

The ability of CBCT to give an image without superimposition was utilized to identify the extent of bone loss, particularly in the buccal and lingual aspects, which are usually too small to be assessed radiographically using the conventional 2-dimensional imaging modalities (periapical or bitewing radiographs). This can give a better understanding to the clinician of why the tooth developed pulpal inflammation or why a tooth with a satisfactory root canal filling and a sound definitive restoration failed and developed apical periodontitis.

Furthermore, it is very important to know the bone levels both buccally and lingually when assessing the crown-root ratio, especially in anterior teeth where the forces exerted are not axial. This becomes valid if there is significant bone loss buccolingually, where the tooth is not as well supported, especially if proximal bone levels do not reflect this amount of bone loss overall. This study has found that crestal bone loss is significantly associated with a higher rate of periapical pathology and the presence of root canal treatment. Periapical lesions are normally of odontogenic origin, and more than 95% of histologic examinations showed either a periapical granuloma or radicular cyst [18]. Nevertheless, a small percentage of periapical lesions might not be a sequel of pulpal necrosis; hence, when periapical pathosis is present, periodontal diagnosis cannot be overlooked [19]. When multi-rooted teeth had periapical pathosis, more than one root tended to be involved. 

Crestal bone loss identified on CBCT was just above 34% of the sample examined, with moderate levels of bone loss being the most common (14.0%). This prevalence of crestal bone loss is similar to many previous studies discussing the prevalence of bone loss [20,21,22]. The pathway of disease progression involves the spread of the microorganisms through the dentinal tubules, lateral and accessory canals, or the apical foramen. Due to the external exposure of these channels leading to the pulp in patients with periodontal bone loss, it is expected that more teeth would present with either inflamed or necrotic pulp, leading to the need for RCT [23,24]. This is reflected in this study, which found a significant relationship between bone loss and RCT. It was shown that 20% of the teeth with periodontal bone loss had undergone an RCT. This finding denotes the potential progression of vital pulp tissue in periodontitis patients to a state requiring RCT. This can guide the clinician in investigating the pulpal status and the need for root canal treatment when patients are diagnosed with periodontal disease. Periodontal therapy, including root planing or other surgical procedures, might also contribute to the exposure of lateral canals and dentinal tubules by removing the cementum layer. 

Many studies showed that defects in cementum or loss of cementum resulting in dentinal exposure would act as a contributing factor to pulpal inflammation and progression towards the need for a root canal treatment. Defects in cementum can arise during root surface debridement or can occur naturally as variations in the relationship between enamel and cementum at the cementoenamel junction [25,26]. An increase in the amount of bone loss does not necessarily lead to more non-vital teeth, which was confirmed in this study as there was no significant statistical relationship between the degree of bone loss and the percentage of teeth with RCT. Premolars were more affected by bone loss than adjacent teeth, potentially due to the presence of deep grooves or root surface irregularities that can impair the periodontal attachment. A study on premolar assessment by CBCT showed that this imaging technique had a good ability to detect root concavities of first premolars and would show the pattern of bone loss associated with the type of concavity [27].

It was found that lateral canals are more prevalent in the apical third, with a percentage of 65% and 70% in the maxillary first premolar and maxillary second premolar, respectively [28]. Regarding molars, lateral canals within the furcation were found to have a prevalence of around 28%, with 25% of molars exhibiting lateral canals limited to the furcation and 10% limited to the lateral wall of the roots [29]. Regardless of the amount of bone loss, in the current study, single-rooted teeth were more likely to have undergone RCT than multi-rooted teeth. This can be explained by the bigger radius at the cementoenamel junction in multi-rooted teeth than in single-rooted teeth; thus, if the rate of bone loss were constant along different teeth, there would be more bone loss in single-rooted teeth, exposing more lateral/accessory canals or dentinal tubules that cause pulpal inflammation and eventually the need for RCT. This study has also shown that a substantial number of the teeth examined had periapical lesions, although no root canal filling was present. This is consistent with other epidemiological studies that have shown an increased prevalence of periapical lesions in the general population. 

Progression of pulp inflammation to an irreversible stage or undergoing necrosis necessitates RCT. If an RCT is not performed or the treatment fails, the progression usually presents as apical bone loss in the form of apical periodontitis [30]. Therefore, it is expected that teeth showing periodontitis-related bone loss would have a higher percentage of apical lesions compared to teeth with healthy bone levels. This finding would emphasize the importance that dentists treating periodontitis patients should test the vitality of teeth and perform RCT where necessary. The presence of apical pathology is associated with a less favorable prognosis for teeth that have had root canal therapy. A systematic review by Ng et al. [31] showed that, for secondary root canal treatment, apical pathosis was associated with a less desirable prognosis. An apical radiolucency was found in almost 40% of root canal treatments when CBCTs were taken in a German population [32]. Inadequate obturation length, taper, presence of voids, and complications were significantly associated with the prognostic periapical status scores [33]. It might be argued that the association might be inverse, i.e., that root-filled teeth are associated with a higher prevalence of bone loss compared to non-root-filled teeth. A few studies have found that adequate root canal fillings in periodontitis patients have no effect on the prognosis of periodontal disease [34].

The results of this study can potentially be used to improve the prognosis for teeth. When a certain degree of bone loss is noted, the clinician can anticipate the need for RCT in the future and closely monitor the tooth to prevent the formation of apical periodontitis, as it has been shown that the development of apical periodontitis adversely affects the success rate of treatment [35]. Furthermore, the approach for periodontal therapy and maintenance should aim to lessen the traumatic impact on the root surface as the dentine-pulpal complex might be exposed due to the loss of cementum. This can involve glycine powder air-polishing (GPAP) in comparison to the mechanical removal of biofilm by root planing. 

CBCT is known to be much more accurate than intra-oral radiographs at detecting furcation lesions and infrabony defects [36]. Due to the higher dose of CBCT than periapical and bitewing radiographs, the routine use of 3D imaging techniques to assess bone levels might not be in line with the principle of ALARA (as low as reasonably achievable) [37]. AAP released a Best Evidence Consensus (BEC), in which the panel of experts concluded that there was presently limited evidence for the routine use of CBCT in the diagnosis and treatment of periodontal disease [36]. However, in a more recent literature review, it was found that there is an increase in the available evidence supporting the use of CBCT in the management of periodontal disease, especially in preoperative planning and follow-up of regenerative surgical treatment, even though it is not recommended as a standard of care [38]. In a systematic review on the use of CBCT in periodontal disease, it was found to be accurate in the management of both infrabony defects and furcation lesions [39]. It can also be justifiable to use CBCT when assessing periodontal diseases around complex structures like maxillary molars [14].

This study was a purely radiographic retrospective study, and there was no clinical component, which is considered a limitation to the study as no clinical investigations were performed to assess the vitality of the teeth. This means that there is a possibility that some teeth have recently become presumably non-vital, and therefore the apical periodontitis has not developed yet. This study examined the presence of apical pathosis at one point in time, and hence the healing/non-healing dynamics of those lesions could not be assessed, and therefore the impact of the presence of periodontal diseases might not be tested reliably.

## 5. Conclusions

CBCT can give the clinician a better insight into the nature of the distribution of crestal bone loss levels around the root apex, localizing the areas that have more bone loss and thus delivering a more accurate diagnosis. This might translate into a treatment plan that is tailored to the geometry of the periodontal defect and a better prognosis. Teeth with crestal bone loss were associated with a significantly higher rate of periapical disease, which highlights the importance of performing a thorough clinical examination in those patients to assess the need for RCTs. Teeth with buccal or lingual bone loss were more likely to have a root canal treatment than teeth with healthy crestal bone levels. Therefore, a comprehensive clinical examination to examine the treatment need for periodontitis patients that includes the assessment of periodontally involved teeth’s vitality is suggested.

## Figures and Tables

**Figure 1 jcm-12-02423-f001:**
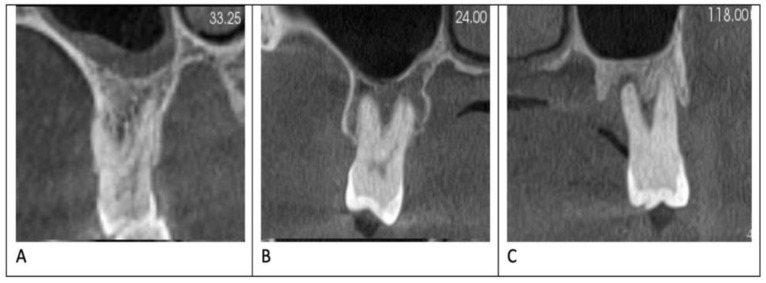
This CBCT cuts demonstrating different degrees of bone loss: (**A**) mild bone loss, (**B**) moderate bone loss, and (**C**) severe bone loss.

**Figure 2 jcm-12-02423-f002:**
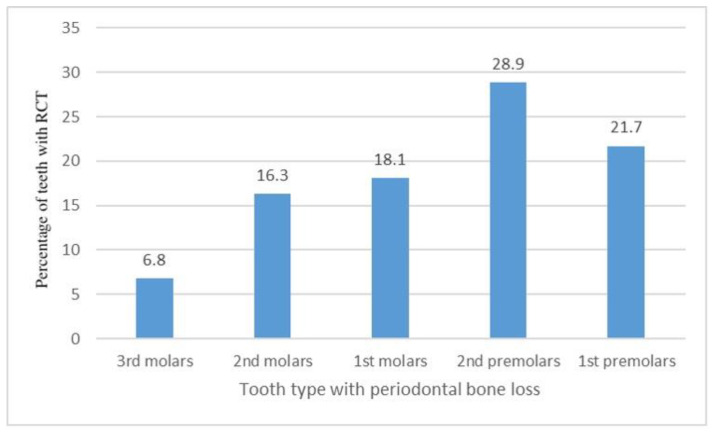
Types of teeth with crestal bone loss and their relation to previous root canal treatments (RCT).

**Table 1 jcm-12-02423-t001:** Endo-periodontal new classification [8].

Endo-Periodontal with Root Damage			Endo-Periodontal without Root Damage					
Root fracture	Perforations	External root resorption	**Periodontitis Patient**			**Non-Periodontitis Patient**		
			Grade 1: narrow, deep periodontal pocket in one tooth surface	Grade 2: wide, deep periodontal pocket in one tooth surface	Grade 3: narrow, deep periodontal pocket in more than one tooth surface	Grade 1: narrow, deep periodontal pocket in one tooth surface	Grade 2: wide, deep periodontal pocket in one tooth surface	Grade 3: narrow, deep periodontal pocket in more than one tooth surface

## Data Availability

Due to the large amount of data extracted, data is available upon request from the corresponding author.

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
