# Peer review of "A Cone Beam CT Study on the Correlation between Crestal Bone Loss and Periapical Disease"

_jcm, 2023, doi:10.3390/jcm12062423_

Round 1

Reviewer 1 Report

Hello, 

An interesting study and well written. 

A few comments: 

1) Periodontitis is a decease diagnosed based on clinical signs and also bone loss mostly on inter proximal. Since this study hasn't looked at either one of the above, I suggest removing any "periodontitis" word especially in the key words and the first paragraph of the results. 

2) In the Method and Material section, the description of the method is deficient. You need to elaborate on the methods used behind selecting the specified cuts of the cross-sectional images and what was the decision based on. I understand that it is challenging to look at the apex and proximal bone on the same cut of the panoramic view on the CBCT. As long as you recognize that the method to standardize the measured cuts must be constant and followed consistently, then it would be acceptable to choose the following cross sectional cut.

3) I suggest rewording of paragraph 6 of the discussion. Too generalized and guess based for a scientific paper. 

4) The message in line 189-190 is very vague. What do you mean by periodontal therapy approach? and is this a suggestion? what is exactly suggested here? 

5) I suggest adding buccal or lingual word to the "crestal bone" because that's what you measured 

Reviewer 2 Report

This paper studied the association of bone loss around dental roots and the peri-apical disease described as radiolucent area that was in contact with the radiographic apex of the root and measured at least twice the width of the periodontal ligament space.  Teeth with crestal bone loss were significantly associated with higher rate of periapical disease and teeth with bone loss were more likely to have a root canal treatment.  This study used a high number of sample size (2001) and demonstrated clear evidence of this association. However, authors in several places in their paper imply that teeth with periapical lesions of described size are non-vital.  This was not shown clinically and it is thus only assumed non-vital.  I would suggest authors refrain from absolute vitality state and use "assumed non-vital"  term instead. 

Reviewer 3 Report

With this manuscript, the authors wanted to confirm that the CBCT method is more effective for determining the level of alveolar bone from all sides of the tooth root, depending on whether the tooth was endodontically treated or not.

Introduction - it is too long and it is necessary to write more about the CBCT method and what and how it serves, as well as its basic characteristics. The data on lesions mentioned in the introductory part should also be shortened. However, the goal was to examine bone loss and CBCT methods.

Material and methods - and this section needs to be reworded...shortened

Results - in the material and methods section, among the parameters, there is demographic data of the patients, but the results do not show for which comparison these data were used.

Discussion - it's too broad...shorten and rewrite.
